# Inclusiveness of Urban Space and Tools for the Assessment of the Quality of Urban Life—A Critical Approach

**DOI:** 10.3390/ijerph18094519

**Published:** 2021-04-24

**Authors:** Agata Gawlak, Magda Matuszewska, Agnieszka Ptak

**Affiliations:** Institute of Architecture, Urban Planning and Heritage Protection, Faculty of Architecture, Poznan University of Technology, 60-965 Poznan, Poland; agata.gawlak@put.poznan.pl (A.G.); magda.matuszewska@put.poznan.pl (M.M.)

**Keywords:** aging societies, quality of urban life, age-friendly design, universal design, inclusive design, tools of assessment, architecture and urban planning

## Abstract

This article aims to compare the international tools assessing the quality of life and to carry out their multifaceted qualitative analysis, emphasizing spatial aspects (architecture, urban planning) and demographic changes. Comparative analysis of three guidelines and 13 rankings includes a comparison of a wide range of domains (2–15), criteria (4–66), and indices (22–223). The already observed worldwide trends of aging societies and increasing urban populations have largely drawn the attention of researchers to the urban life quality. Since the early 1990s, many international tools have been developed for that purpose. Although urban practitioners progressively rely on instruments measuring urban quality of life, in the development of urban policy, there is still little research comparing the already available appraisal instruments in view of their measurement criteria. The results of the research on the global tools show that there are major differences between them, either in view of the purpose, the contracting authorities, research focus group, scale, or in view of the importance of spatial and demographic factors. Such findings can contribute to the development of local guidelines and recommendations for self-government authorities and communities, in this the seniors and future generations, in view of improving the urban life quality.

## 1. Introduction

We can observe major changes worldwide in the demographic population structure connected with the aging society phenomenon. According to the UN Department of Economic and Social Affairs, the population in the age from 0 to 24 will be going down, and the share of the age group of 65+ will become larger and larger. This tendency is clearly visible in Europe due to a significant increase in the average life span. The percentage of people aged 65+ has increased from 21.8% in 2000 to 29.5% in 2020 in Europe, whereas in Africa, this increase has amounted to 0.1 of the percentage point. In Italy, the country in Europe characterized by having the longest life span, the percentage share of seniors aged 65+ has significantly increased from 27.1% in 2000 to 36.6% in 2020. Moreover, the urban population is also rapidly increasing. The beginnings of the trend date back to 1950 when the urban population was 751 million. In 2018 this number surged to 4.2 billion, which represented 55% of the global population. According to the forecasts, this number will have increased to 6.7 billion in 2050, which will correspond to over 2/3 of the global population (68%) [1].

Since the early 1990s, the aspects of the quality of life started to attract more and more interest, caused by a growth in international consideration of the environmental and urban issues [2,3]. A great number of works was published on this topic [4]. International organizations and bodies, government and research units, businesses, or publishing entities began to measure the quality of urban life with the use of assessment tools based on selected criteria and indicators, often relying on the comparisons of various factors. These include societies, towns, and countries, which then make up the bases for the rankings and ratings of the best and worst places to live [5,6,7,8,9,10,11,12,13,14,15,16,17].

“*Places Rated Almanac*” has been recognized as the first urban ranking that influenced popularization of the rankings concerning aspects of the quality of life in a city [2,18,19]. However, it has been the overall wider notice of the system of indicators that caused a total increase in community and managers’ attention paid to rankings [2,20].

“As Robinson (2016) has pointed out, ‘comparative tactics’ can have a crucial place in advocating for ‘a more global urban studies’ where a wider set of urban experiences are included. This is not just a role of advocating for more cities to be included. It is also a deeper methodological and empirical contribution to advocate for better measurements and benchmarks that capture the variety of factors that shape cities and the different types of urban dwellers that inhabit them beyond the common creative class viewpoints” [2,21].

The researchers have also proposed relevant recommendations of a varying degree of detail, concerning the assessment of the urban life quality, with particular account for the seniors [22,23,24]. Apart from the aforementioned tools, there are also guidelines that highlight the city’s performance not generally but in one particular domain, e.g., streets. The human-centered framework The *Healthy Streets Approach* developed by Lucy Saunders aims to combine public health and urban spaces and verify the human experience of being on the streets on the basis of 10 evidence-based indicators [25,26]. Another tool worth mentioning is the *Urban Street Design Guide* from Nacto (National Association of City Transportation Officials). This design guide emphasizes the role of streets in public life, taking into account not only the current needs but also the future challenges, which, in comparison to other tools, may be concerned as a new approach [27].

The tools that are available to us today differ so much that it seems fully justified to carry out their comparative analysis and to assess the importance of the domains and criteria they apply. There is no published research comparing the aforementioned features of the diverse tools used for the assessment of the quality of urban life. Even if the article entitled *“A Critical Review on the Worldwide Economist Intelligence Unit, Mercer and Monocle Quality of Life Indicators”* describes a comparative analysis of the rankings, it is limited just to 3 of them, of which all were contracted for commercial purposes and were addressed to a similar target group of inhabitants of the most representative cities. The authors of this article have compared the criteria used in all the rankings analyzed herein, accounting for five selected domains (politics, economics, social environment, and infrastructure) [28].

A multifaceted analysis of various international tools used to assess the quality of urban life may prove useful, especially if these are focused on the selected domains and the assessment criteria that are significant in view of the quality of life of inhabitants of a given city/town. Such tools can then underlie the development of guidelines and evaluation methods, not only on the global scale but also on the local scale.

All the tools refer to the aspects connected with the quality of life, whereas their large diversification has forced us to pose the following questions:Are we able to come up with an unambiguous definition of urban life quality?How much are spatial aspects important in the research on the quality of urban life?Have all the rankings taken into account the demographic aspects?

## 2. Materials and Methods

The basis for this multifaceted analysis of various international tools used to assess the quality of urban life was the studies of the reference literature. They represented an overview of the current state of knowledge on the issue of the quality of urban life and on the available guidelines and tools for the assessment of the quality of life and aging societies. In view of architectural design, the assessment criteria included such aspects as accessibility of housing and healthcare, universal and inclusive design. Based on the studies of the reference literature and the accessibility criterion, 16 prestigious and influential global tools (oriented to evaluate or amplify the concept of the quality of urban life) were selected (Figure 1). The multifaceted analysis included 3 sets of guidelines [22,23,24] and 13 ratings [5,6,7,8,9,10,11,12,13,14,15,16,17], of which 4 contracted for commercial purposes by *The Economist Intelligence Unit* [5], *Mercer* [6], *Magazine Monocle* [7], *Deutsche Bank AG/London* [8], and 12 by non-commercial institutions, that all concern livable cities’ features and indicators for the quality of urban life assessment (Table 1).

The comparative analysis of the selected tools included the qualitative analysis of the assessment method. The research was carried out accounting for a number of aspects, such as: the contracting authorities, the main purpose of the research, a sample size (Table 2), a research focus group, the intended audience of the studies, or criteria used in a given tool to assess the quality of life (Table 2).

On the basis of the comparative analysis of respective criteria and indicators included therein, data synthesis was conducted, and 7 domains (areas of interest) were proposed, of which 6 main domains and 1 domain of “supplementary criteria” deemed as auxiliary for the purpose hereof by the author. All these domains have underlain a joint and universal classification of the criteria used in all of the analyzed tools (Table 3). The report *Global Age-friendly Cities—A Guide*, being the most extensive assessment tool, was selected as a source of reference [16].

The comparative analysis was intended to show the share of spatial and demographic aspects in the urban life quality assessments. The multifaceted analysis was additionally based on the statistical analysis of the cities/towns assessed depending on the parameters related to a given city/town size (population) and to their administrative and political function. Finally, the criteria were mapped, and the coverage matrix was presented, broken down by respective criteria and their share in the assessment (Figure 2).

## 3. Results

The multifaceted qualitative analysis of the guidelines and rankings/ratings of urban life quality will result in new knowledge on the available assessment tools.

All the tools, despite their correlation with the studies on the quality of life, are characterized by their own individual approach to the topic. Depending on the tool, the assessed life quality is referred to the perception of quality, living standard, technology, or the health care system (Table 5).

The research has identified differences in such aspects as the main purpose, the inclusion of the demographic factor in the assessment (Table 4), sample size (Table 2), selection of the research focus group, the intended audience of the studies, and the contracting bodies (Table 6). It has, furthermore, revealed that the number of the proposed domains and indicators (Table 2) and the weight of domains and criteria concerning the spatial aspects (Figure 3 and Figure 4) differ between the tools.

The studies show that there are no uniform parameters and that the assessments include many soft, objective, and subjective aspects. The ratio of objective and subjective indicators is also different. Some of the tools include only the objective aspects related to the living standard and some—only the subjective aspects related to the perception of quality. Only three tools: Quality of life (well-being of Europeans) [13], The European Quality of Life Survey [14], and How’s Life? 2020 Measuring Well-being [15] include both objective and subjective aspects in their assessments.

### 3.1. Assessment Domains and Criteria Related to Spatial Aspects

The way we shape space has an undeniable impact on the quality of urban life. Cities that adhere to the principles of universal and inclusive design are adapted to the needs of all their inhabitants and positively influence many spheres of urban life. Properly arranged space can facilitate social interactions, increase educational chances and employment opportunities, affect the feeling of safety and bring the urban dweller closer to nature. Any assessment of the quality of life should account for the spatial aspects, inclusive of the private space, residential space, urban space, or the space in public utility buildings.

Our research has analyzed the share of the domain called “architecture and urbanism” and the shares of the other domains assessed within the framework of various tools as influencing the quality of urban life. As regards the international guidelines, the domain of “infrastructure” represents the largest share (26.3%). According to the bodies contracting various rankings, “health and well-being” has the highest weighted average in the assessment of the urban life quality (24.2%). In the same tools, the weighted average of the “architecture and urbanism” domain in the urban life quality assessment is the lowest (4.3%) (Figure 3).

Respective shares of the criteria included in the “architecture and urbanism” domain were further analyzed, too. Their shares in the guidelines were as follows: “housing”—10.3%, “public buildings”—5.5%, and “outdoor spaces”—7.6%. The “housing” criterion was also deemed to be the most important in the rankings—3.3%, whereas the remaining criteria of the “architecture and urbanism” domain scored only 0.3% (“public buildings”) and 0.7% (“outdoor spaces”) of the weighted average (Figure 4). The criterion that is most often included in the life quality assessment tool is the “housing” criterion (81.3% of the analyzed tools), and the most rarely used criterion is “public buildings” (only 31.3% of the analyzed tools) (Figure 2).

The tool that assures the largest range of criteria and their weight in respect to the analysis of the impact of space are the guidelines of WHO Global Age-friendly Cities—A Guide (1.1) [22]. Three international rankings: Euro Health Consumer Index (2.5) [9], IMD Smart City Index (2.6) [10], and Active Ageing Index (2.13) [17] entirely excluded the spatial aspect from the assessment (Figure 2).

### 3.2. International Rankings

Half of the analyzed rankings examine and compares the situation in the selected European countries, OECD and UN countries, and the other half focuses on the analysis of the most representative cities in the world. The analyzed cities are most often the capital cities of the countries, regions, provinces, business, economic, cultural, or university centers, or the most densely populated cities. The Best Cities for Successful Aging ranking is an exception. It analyses 381 metropolitan districts in the USA [11]. Most of the rankings are targeted at the government authorities, social policy activists and leaders, leaders of the countries or municipal officials. Three rankings: Best Cities for Successful Aging [11], Quality of life (well-being of Europeans) [13], and Quality of life in cities. Perception surveys in 79 European cities [14] also name the citizens as their recipients. The results of the analysis conducted by the Monocle magazine are also intended to be used by architects, urban planners, and financial analysts [7]. Euro Health Consumer Index is addressed to the institutions liable for the organization of health care in a given country [9] and Deutsche Bank Liveability Survey to global financial markets [8].

### 3.3. International Guidelines

All the analyzed international guidelines include a list of features of a given place (city/town, rural area) that is senior-friendly. The list can be further used for the self-assessment of given locations and for the verification of whether the features are available to the seniors. *The Global Age-friendly Cities—A Guide* guidelines take into account the largest number of indicators (168) (Table 2), which are allocated to eight different domains (outdoor spaces and buildings, transportation, housing, social participation, respect and social inclusion, civic participation and employment, communication and information, community support and health services). This assessment project has also surveyed the largest group of respondents (1485). The respondents were divided into 158 research focus groups and came from 33 cities in the world. In all the analyzed guidelines, seniors were the research focus group [22,23,24]. Carers of the disabled and providers of services dedicated to the seniors took part in two of the research projects. Experts and local authorities were involved only in one research study. The guidelines are addressed to all the parties interested in increased accessibility of certain urban life aspects to the seniors, i.e., to the municipal authorities, self-governments, volunteer organizations, the private business sector, and the seniors themselves. The size of the population samples differed. All three guidelines: *Global Age-friendly Cities—A Guide, Measuring the age-friendliness of cities. A guide to using core indicators, Age-friendly rural and remote communities: a guide* included a detailed survey of the seniors and took account of the demographic changes (Table 4) [22,23,24].

## 4. Discussion

Due to the continuing demographic changes and the extending life span, the seniors shall be accounted for, in the research, in particular. Most of the rankings excluded any aspects related to the aging society trends. The issue of the aging society was especially considered in the guidelines. These were developed in order to study the accessibility of the cities/towns to the seniors [22,23,24]. In addition, two rankings: *Best Cities for Successful Aging* and *Active Ageing Index,* focused on a similar topic (Table 4) [11,17].

The aspect most worth considering is the one underlying the selection of particular towns/cities for the assessment. According to the authors of *WHO Global Age-friendly Cities: A Guide*, the selected cities were the representative cities of developed and developing countries, diverse in view of the urban tissue, size, and population. At the assumed criterion of a small town characterized with the population ranging from 1000 to 10,000 inhabitants. Only one town was classified for the research project as meeting that criterion, and so were only four towns of the population ranging from 10,000 to 100,000 inhabitants, out of the total of 33 selected cities/towns [22]. Towns/cities researched within the framework of *Measuring the age-friendliness of cities. A guide to using core indicators* were defined as a diverse group of pilot cities/towns. The selection process accounted for the number of inhabitants, area size, and tissue (urban, rural). On top of this, the membership in the Global Network of Age-friendly Cities and Communities, the social and cultural context were also considered as important criteria. From among the towns with the population ranging from 1000 to 10,000 inhabitants, two towns were classified for the research project as meeting that criterion, and two of the population ranging from 10,000 to 100,000 inhabitants, out of the total of 15 selected cities/towns [23].

The analyzed tools differ in view of their purpose. The tools were proposed to be divided according to their relevant prevailing purpose. The analysis of aspects that make the towns/cities friendly for their aging inhabitants was the prevailing purpose of all the analyzed international guidelines and two rankings: *Best Cities for Successful Ageing* and *Active Ageing Index* [9,15,16,17,18]. Rankings contracted by commercial entities mainly surveyed the aspects related to the living standard. The ranking produced by the *Monocle* magazine stands out in that category because it included additional subjective indicators related to the happiness and comfort felt in daily life [7]. The life quality assessment, inclusive of objective and subjective aspects, was a prevailing goal of the studies conducted by *Eurostat (Quality of life)* and *OECD (How’s life? 2020 Measuring Well-being)* [13,15]. The main aim of the research carried out by *the European Commission (Quality of life in cities. Perception survey in 79 European cities)* was the examination of the perception of quality. Other tools assessed the health care system, infrastructure and technology, and “human development” (Table 5) [9,10,12].

The types of contracting authorities turned out diverse, starting from research institutes and universities and colleges, through financial and business institutions, international bodies, and organizations, ending on representatives of the publishing industry (Table 6).

Four rankings: EIU’s Global Liveability Ranking, Mercer’s Quality of Living Ranking, Monocle’s Quality of Living Survey, and Deutsche Bank Liveability Survey were contracted by commercial entities; this fact largely affected the selection of the test group and the results recipients group. The sample of cities/towns selected for the research (Table 2) ranged from 56 (Deutsche Bank Liveability Survey) to 393 (Mercer’s Quality of Living Ranking). All of them were large and representative urban agglomerations. The majority of the cities/towns classified for the rankings were situated in Asian or European countries. The minority were in South America and Oceania. The intended audience of the analyses were the national governments, international companies, businessmen, businesswomen, global financial markets, and persons planning to seek employment in a given city/town [5,6,7,8].

The definition of the term “life quality” was verified, and so was the importance of the spatial aspects and inclusion of the demographic changes in the research. The analysis of different tools of assessment shows that there is no unambiguous definition of the quality of urban life, and there are no uniform methods of its assessment; there are many soft, objective, and subjective aspects included. The differences in using quantitative and/or qualitative measures in appraisal instruments have also been confirmed by other researchers [2].

Despite the fact that all these assessment tools try to study the issue of the quality of life, they largely differ in view of the size of the test group, the selection process, the prevailing goal of the analysis attempting to grasp the inherent components of life quality, the intended audience of the study and focus groups. A significant factor is the fact that the majority of the analyzed rankings fail to account for the demographic aspects connected with the aging of societies [5,6,7,8,9,10,12,13,16] (Table 4). Lack of any criteria or even domains related to the spatial aspects of life quality in some of the analyzed tools is equally important (Table 2). The analysis of shares of respective criteria shows that their contribution into the same domain can be differently interpreted (Figure 4).

The research was limited to the analysis of the selected tools of assessment. This might adversely affect the objectivity of the results. Nevertheless, the results justify the statement that a selected group of assessment tools is founded on schemes. For example, it is worth underlining that the tools that take into account the demographic changes represent tools of a defined profile dedicated to the studies on the accessibility of the seniors to a variety of urban locations. The international tools most often compare large and representative cities, and the type of contracting authority is closely related to the prevailing purpose of the assessment.

The qualitative analyses show that the available global tools of assessment are usually adapted to the realities of large cities, which perform high administrative and political functions, which impedes the option of their adjustment to the realities of low ranking small towns. The guidelines *Age-friendly rural and remote communities: a guide* make up an exception here as they account for an option of smaller scale towns [24]. The researched groups of towns are non-homogeneous, whereas the tools are applied globally. According to urban researchers, rankings are also decontextualizing a city [29] and implying the city as a closed, individual system [30], although it is involved in the urban network or system that originate from the interactions between particular cities [31]. Taylor claims that “data on cities are available in state census reports and this has allowed for comparative city research but there is nothing on relations between cities across state boundaries.” The available information enabling comparative analyses of the world cities overall remains very limited, as most of the published data is formed by states to achieve only state goals. Hence, the result of ranking is an ordered list not providing any information about the connection between the ranked cities [32].

Therefore, comparisons between the cities are further impeded by different cultural and social contexts. Aspects concerning religion are considered only in a few of the tools. They are usually included as sub-points in the indicators connected with the society, culture, and architecture, e.g., with religious restrictions, freedom of religious beliefs, or close distance from the place of residence to the sacral buildings. Only two rankings analyzed the issue of religious denomination as a separate index (2.7—“number of civic and religious organizations”; 2.10—“tension between different religious groups”, “attendance at religious services apart from weddings, funerals, christenings”). The absence of any indicators related to religion in rankings contracted by the *Economist, Mercer,* and *Monocle* was pointed out in the work of Sarkawi, Abdullah, and Dali [28].


*“Hence it has been observed that the most striking indicator found absent in all the liveability indicators by the world organizations is the specific religious indicator”*
[28] (p. 590)

This analysis of the inclusion of the cultural context in comparable tools of assessment showed that most of the tools had marginalized that aspect. *The Global Age-friendly Cities—A Guide* and *Measuring the age-friendliness of cities. A guide to using core indicators* guidelines propose a set of universal criteria and indicators, which should next be adjusted to the local context [22]. *The Age-friendly rural and remote communities: a guide* guidelines refer to small communities in Canada. This document underlines the fact that the researched communities largely differ in view of the cultural background, distance of the location, the nearest city, etc. The guidelines further state that the proposed solutions need to be reviewed by the communities individually, accounting for their local conditions [24]. The research should, however, include the cultural context, analyzed in-depth through the prism of the conditions of local communities. Therefore, two components are inherent in the concept of the quality of life: the objective one that refers to the living standards and the subjective one that refers to the feeling of general satisfaction and happiness with respect to various aspects of life [4]. However, the global tools (rankings) do not sufficiently take into account the cultural context, even if they do include indicators for the comparisons of various countries (in these different towns and communities located within their borderlines), and their questions cover the objective and subjective assessment aspects. At the same time, the other tools that focus on the analyses of the cities *EIU’s Global Liveability Ranking, Mercer’s Quality of Living Ranking, Deutsche Bank Liveability Survey,* or of the metropolises *Best Cities for Successful Aging* fail to include any subjective indicators. The authors of *Monocle’s Quality of Life Survey* classify some of the indicators as subjective, but it must be observed that these pertain to the experts’ opinions rather than the perception of the inhabitants. Whereas, *IMD Smart City Index* and *Quality of life in cities. Perception survey in 79 European cities* rankings entirely neglect the objective aspects, focusing only on the subjective assessment of the inhabitants.

## 5. Conclusions

We are currently facing changes related to aging societies and increasing urban populations. Thus, we require the urban tissue to be properly shaped to be able to adequately respond to the changes. Universal and inclusive space can motivate communities to a more active lifestyle and positively affect the health of its dwellers. Spatial aspects are, without any doubt, related to other domains of the quality of life and, for that reason, their share therein should be increased. Perception of the quality of life depends on age. Young people and seniors have different needs and priorities. That is why a multifaceted analysis should address all the needs of a variety of members of the researched communities.

It is argued that urban rankings assessing the quality of life can be potentially used in the development, as well as the creation of urban policy [2,3,33,34,35]. Urban practitioners can use rankings to evaluate the present state in order to compare it to the situation of other cities and learn about their policy strategies. In consequence, it leads to creating collaborative networks [2].

According to the researchers, “there is also a critical role for academia in developing and improving methodologies for city benchmarking and making these publicly available. Whilst we have highlighted some examples of scholarled benchmarking studies, these remain by far the minority, and in the cases where scholars have developed studies on behalf of private companies, a detailed methodology is often not reported” [2,33].

The research has resulted in a multifaceted qualitative analysis of the available tools of assessment of urban life quality. The limitations mainly concern data availability and accessibility. In some cases, there is a lack of transparency in the methodology description and definition of particular indicators. The aforementioned limitations are also assigned to the rankings themselves in the scientific literature [2].

As the number of indicators concerning spatial aspects needs to be extended, the domains and criteria aforementioned in the results can be used for developing appraisal instruments.

It can, further, serve as the basis for a conscious decision-making process in local government offices in view of the specific needs of particular regions. Adaptation of global tools to the specific local conditions and context can prove useful in solving local issues and can underlie the development of the guidelines and tools of assessment adapted to the specifics of smaller towns. It shall, in particular, focus on the needs of seniors regarding public space and housing. The tools in question shall be adequately adapted to their purpose, scale, and context because the differences between the instruments (guidelines/rankings)—identified and proven herein—are significant, and therefore, they cannot be confirmed to have a universal application.

## Figures and Tables

**Figure 1 ijerph-18-04519-f001:**
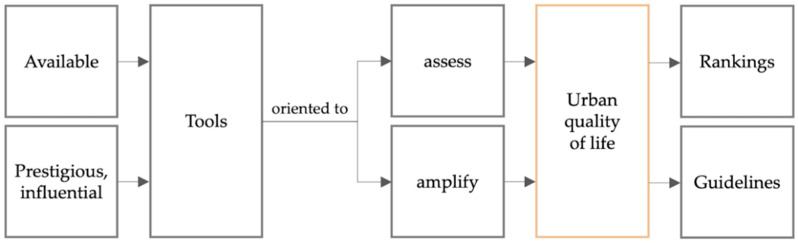
Scheme on the research design: criteria applied for the tools selection.

**Figure 2 ijerph-18-04519-f002:**
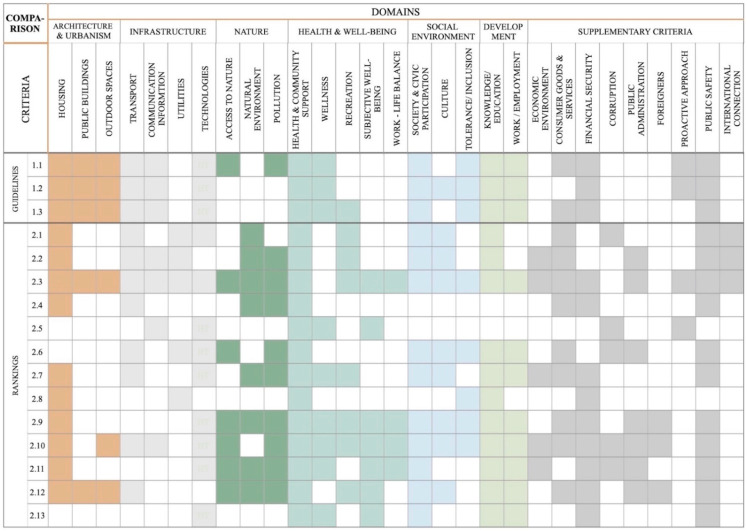
Share of respective criteria in the assessment: coverage matrix.

**Figure 3 ijerph-18-04519-f003:**
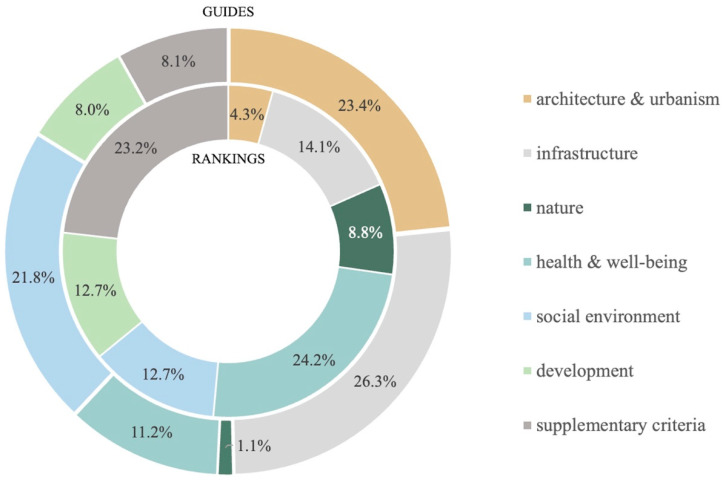
Share of the “architecture and urbanism” domain in reference to other domains taken into account in the guidelines and rankings.

**Figure 4 ijerph-18-04519-f004:**
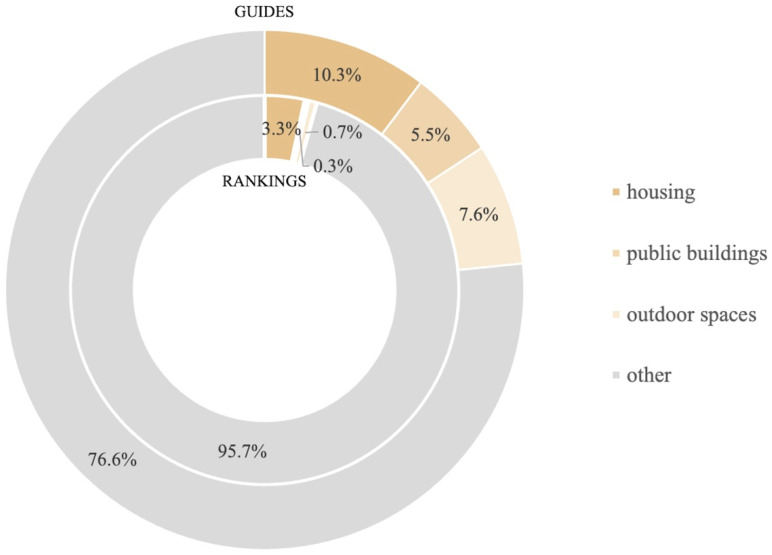
Share of the “housing”, “public buildings”, and “outdoor spaces” criteria in reference to other criteria used in the guidelines and rankings.

**Table 1 ijerph-18-04519-t001:** Selected appraisal instruments: guides and rankings. Contracting authorities.

Table	NO.	Name	Contracting Authority
GUIDES	1.1	Global Age-friendly Cities—A Guide [22]	World Health Organization
	1.2	Measuring the age-friendliness of cities. A guide to using core indicators [23]	World Health Organization
	1.3	Age-friendly rural and remote communities: a guide [24]	Federal/Provincial/Territorial Ministers Responsible for Seniors
RANKINGS	2.1	EIU’s Global Liveability Ranking [5]	The Economist Intelligence Unit
	2.2	Mercer’s Quality of Living Ranking [6]	Mercer
	2.3	Monocle’s Quality of Life Survey [7]	Monocle
	2.4	Deutsche Bank Liveability Survey [8]	Deutsche Bank AG/London
	2.5	Euro Health Consumer Index [9]	Health Consumer Powerhouse
	2.6	IMD Smart City Index [10]	IMD World Competitiveness Center’s Smart City Observatory Singapore University of Technology and Design (SUTD)
	2.7	Best Cities for Successful Aging [11]	Milken Institute Center for the Future of Aging
	2.8	Human Development Report [12]	United Nations Development Programme (UNDP)
	2.9	Quality of life (well-being of Europeans) [13]	Eurostat, European Commission
	2.10	The European Quality of Life Survey [14]	The European Foundation for the Improvement of Living and Working Conditions (Eurofound)
	2.11	How’s Life? 2020 Measuring Well-being [15]	OECD
	2.12	Quality of life in cities. Perception survey in 79 European cities [16]	European Commission
	2.13	Active Ageing Index [17]	UNECE and the European Commission

**Table 2 ijerph-18-04519-t002:** Tools. Quantitative data: number of cities, countries, criteria, and indices.

Table	No. of Cities	No. of Countries	No. of Domains	No. of Criteria	No. of Indices
GUIDES	10–33 (19)	1–23 (12)	4–8 (7)	22–66 (45)	43–169 (110)
RANKINGS	56–393 (169)	28–189 (63)	2–15 (7)	4–39 (20)	22–223 (71)

**Table 3 ijerph-18-04519-t003:** Criteria classification domains: authors proposal.

1.	Architecture and urbanism
2.	Infrastructure
3.	Nature
4.	Health and well-being
5.	Social environment
6.	Development
7.	Supplementary criteria

**Table 4 ijerph-18-04519-t004:** Classification of tools. Aspect of an aging society.

Concerning Older People Aspect	Global Age-friendly Cities—A Guide
	Measuring the age-friendliness of cities. A guide to using core indicators
	Age-friendly rural and remote communities: a guide
	Best Cities for Successful Aging
	Active Ageing Index
Partially Concerning Older People Aspect	The European Quality of Life Survey (EQLS)
	How’s Life? 2020 Measuring Well-being
No Older People Aspects	EIU’s Global Liveability Ranking
	Mercer’s Quality of Living Ranking
	Monocle’s Quality of Life Survey
	Deutsche Bank Liveability Survey
	Euro Health Consumer Index
	IMD Smart City Index
	Human Development Report
	Quality of life (well-being of Europeans)
	Quality of life in cities. Perception survey in 79 European cities

**Table 5 ijerph-18-04519-t005:** Classification of tools. The strategic purpose.

Age-Friendliness	Global Age-friendly Cities—A Guide
	Measuring the age-friendliness of cities. A guide to using core indicators
	Age-friendly rural and remote communities: a guide
	Best Cities for Successful Aging
	Active Ageing Index
Standard Of Living	Mercer’s Quality of Living Ranking
	Deutsche Bank Liveability Survey
	EIU’s Global Liveability Ranking
Quality of Life	Monocle’s Quality of Life Survey
	The European Quality of Life Survey (EQLS)
	How’s Life? 2020 Measuring Well-being
	Quality of life (well-being of Europeans)
Quality in People’s Perspective	Quality of life in cities. Perception survey in 79 European cities
Infrastructure and Technology	IMD Smart City Index
Human Development	Human Development Report
Healthcare System	Euro Health Consumer Index

**Table 6 ijerph-18-04519-t006:** Classification of tools. Type of contracting authorities.

Research Institute	Euro Health Consumer Index
	Best Cities for Successful Aging
University/College	IMD Smart City Index
Financial Institution	Deutsche Bank Liveability Survey
International Body	Age-friendly rural and remote communities: a guide
	Quality of life (well-being of Europeans)
	The European Quality of Life Survey (EQLS)
	Quality of life in cities. Perception survey in 79 European cities
	Active Ageing Index
International Organization	Global Age-friendly Cities—A Guide
	Measuring the age-friendliness of cities. A guide to using core indicators
	Human Development Report
	How’s Life? 2020 Measuring Well-being
Business Institution	EIU’s Global Liveability Ranking
	Mercer’s Quality of Living Ranking
Publishing Industry Sector	Monocle’s Quality of Life Survey

## Data Availability

The data that support the findings of this study are available from the corresponding author, AP, upon reasonable request.

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
