# Peer review of "Inclusiveness of Urban Space and Tools for the Assessment of the Quality of Urban Life—A Critical Approach"

_ijerph, 2021, doi:10.3390/ijerph18094519_

Round 1
Reviewer 1 Report
The paper focuses a relevant domain for urban strategic development: aging dynamics.
The paper is based on a qualitative assessment comparing main domains included in two different categories of tools oriented to assess or make explicit the concept of "urban quality of life": rankings & guidelines.
Analysed sources are relevant and heterogeneous.
Apart from discussing differences between methods and the classification according to the relevant domains proposed by the authors, I have to point out that a prosal on a better combination of such domains/variables could be asy included in this work. In fact the reader actually misses a proposal deriving from this interesting analysis.
I believe that this effort can be request to the authors in the perspective of a significative improvement of the research providing a position or, in other words, a more explicit conclusion for the analytical approach.
That's why I suggest major revision.
Concerning bibliography, I suggest to consider relevant authors for methodological approach to cities ranking, urban development, urban quality. It could improve the discussion better linking the results to the concept of urban quality of life.
some suggestions:
Taylor, P. J. (1997). Hierarchical tendencies amongst world cities: A global research proposal. Cities, 14(6), 323–332. https://doi.org/10.1016/s0264-2751(97)00023-1
Acuto, M., Pejic, D., & Briggs, J. (2021). TAKING CITY RANKINGS SERIOUSLY: Engaging with Benchmarking Practices in Global Urbanism. International Journal of Urban and Regional Research, 45(2), 1468-2427.12974. https://doi.org/10.1111/1468-2427.12974
Batty, M. (2006). Rank clocks. Nature, 444(7119), 592–596. https://doi.org/10.1038/nature05302
Pumain, D. (2006). Hierarchy in Natural and Social Sciences.
Waiting to read the revised version, compliments for selecting this relevant topic that should rais the agenda of decision makers concerning sustainable urban development.
Author Response
Dear Reviewer,
Thank you for taking the time to assess our manuscript. We appreciate your insightful comments and excellent suggestions. They have been very helpful in improving the manuscript. The references that you suggested helped us develop the theoretical framing of the study, for which we are very grateful. Please see the attachment, with a point-by-point response, in red, to your comments and concerns.
Yours sincerely,
Agnieszka Ptak

Reviewer 2 Report
The paper is interesting and well structured. The general applied methodology is innovative, related to monitoring, evaluating, measuring and assessing the the quality of urban life.
The abstract could benefit from two or three sentences at the beginning to introduce the overall problem statement. The introduction provides a great overview of the problem under investigation, including the methods aspects. The discussion is strong but often times it gets too descriptive.
Overall, the text is well developed and allows an easy understanding of the issues dealt with also with the help of the tables that well represent the study carried out.
Please check the links in the references because some web sites have been moved or deleted (for example, the reference no. 16)
In a possible continuation of the research for the purposes of guidelines, it is recommended to expand the case studies with the Healthy Streets Approach to Urban Planning developed by Lucy Saunders, a framework for putting human health and quality of life at the center of decision making around transport and public realm planning and management. Another useful reference could be the Urban Street Design Guide from Nacto (National Association of City Transportation Officials)
Author Response
Dear Reviewer,
Thank you for taking the time to assess our manuscript. We highly appreciate all the valuable and insightful comments. They have been very helpful in improving the manuscript. The changes are highlighted within the manuscript. Please see the attachment, with a point-by-point response, in red, to your comments and concerns.
Yours sincerely,
Agnieszka Ptak

Reviewer 3 Report
Dear editors of IJERPH,
thank you for the possibility for review an interesting manuscript. My observations and comment can be found bellow:
- Abstract. I think I understood what it the research presented in the manuscript about, however, more clarity in the abstract is needed. As the readers usually read the abstracts independently without any further knowledge about the research conducted, the information presented needs to be concise and well structured. I think that the abstract, as it is now, is very general and rather vague. I think that we understood what is the aim but we need to know more about the methodology and even more about the results. The presented research are too general and obvious and expectable. Is it please possible to highlight here some more from the results? Please consider also highlighting the uniqueness of the study. Is it also please possible to say more about practical usage of the comparison of tool? English needs to be more advanced.
- Title. The title could be more informative to be more in line with the content. Maybe adding the key research question to the title would be helpful.
- Introduction. The introduction serves here also as a theoretical framing of the study which is, unfortunately, lacking here. We truly need to know what theories are the authors working with and what are the current frontiers in the field. This cannot be understood from the Introduction. Could you please expand this part? Strong statements should be always supported by referencing as these make the statements stronger. The research question should be based on the findings from previous studies and the identification of a research gap. This is not clear in this case.
- Methodology. Also here we need to better comprehend individual phases of the research. We need to better understand how the tools were selected. Was some criteria applied? It would be brilliant if a scheme on the research design could be shown in this part. Please check the titles of figure, some of them seem to be too generic.
- In table 1 more information about the individual tools should be provided. We should know the objectives of the tools, about the data these are working with and how the tools are designed, etc.
- In the methodology, please focus more on an in-depth description of every phase of the research.
- The results seem reasonable but my overall feeling is that the results and somehow mixed here altogether with the parts that should rather be, in my opinion, the part of the discussion. Please try to more develop your results and interpret them.
- I really do like the graphics used in the section on results. Are there some more possibilities how the findings could be graphically shown?
- Please include the limitations of the study in the concluding part. Are there any limits concerning the usage of the methods and data?
- The list of references need to be expanded to reflect the most recent advances in the field.
I think that the paper will be a good fit for the journal when revised. As the moment, the paper cannot be accepted for publication and some work still has to be done. Please focus on theoretical framing of the study and on interpretations of the results. Please consider English edits by a native speaker.
Many thanks for the possibility to review this paper.
I recommend a major revision.
Kind regards,
Author Response
Dear Reviewer,
Thank you for taking the time to assess our manuscript. We greatly appreciate your valuable and insightful comments. We have carefully considered all of them and made the suggested modifications in the manuscript. We believe that the manuscript has been improved. The changes are highlighted within the manuscript. Please find the attachment, with a point-by-point response, in red, to your comments and concerns.
Yours sincerely,
Agnieszka Ptak

Round 2
Reviewer 3 Report
Dear editors,
many thanks for sharing the new version of the paper with me. After careful reading, I am very happy to say that all my comments were sufficiently reflected in the paper and from my perspective, the paper is much more internationally competitive now. I think that it might be accepted.
Let me congratulate the authors for their excellent research presented here.
Kind regards,